# Rapid Liquid Chromatography—Tandem Mass Spectrometry Analysis of Two Urinary Oxidative Stress Biomarkers: 8-oxodG and 8-isoprostane

**DOI:** 10.3390/antiox10010038

**Published:** 2020-12-31

**Authors:** Nicolas Sambiagio, Jean-Jacques Sauvain, Aurélie Berthet, Reto Auer, Anna Schoeni, Nancy B. Hopf

**Affiliations:** 1Center for Primary Care and Public Health (Unisanté), University of Lausanne, Route de la Corniche 2, 1066 Epalinges-Lausanne, Switzerland; nicolas.sambiagio@unisante.ch (N.S.); jean-jacques.sauvain@unisante.ch (J.-J.S.); aurelie.berthet@unisante.ch (A.B.); reto.auer@biham.unibe.ch (R.A.); 2Institute of Primary Health Care (BIHAM), University of Bern, Mittelstrasse 43, 3012 Bern, Switzerland; anna.schoeni@biham.unibe.ch

**Keywords:** oxidative stress, biomarker, 8-oxodG, 8-isoprostane, biomonitoring, liquid chromatography, mass spectrometry

## Abstract

Human biomonitoring of oxidative stress relies on urinary effect biomarkers such as 8-oxo-7,8-dihydro-2′-deoxyguanosine (8-oxodG), and 8-iso-prostaglandin F2α (8-isoprostane); however, their levels reported for similar populations are inconsistent in the scientific literature. One of the reasons is the multitude of analytical methods with varying degrees of selectivity used to quantify these biomarkers. Single-analyte methods are often used, requiring multiple injections that increase both time and cost. We developed a rapid ultra-high-performance liquid chromatography–tandem mass spectrometry (UPLC-MS/MS) method to quantify both urinary biomarkers simultaneously. A reversed-phase column using a gradient consisting of 0.1% acetic acid in water and 0.1% acetic acid in methanol/acetonitrile (70:30) was used for separation. The MS detection was by positive (8-oxodG) and negative (8-isoprostane) ion-mode by multiple reaction monitoring. Very low limit of detection (<20 pg/mL), excellent linearity (R2 > 0.999), accuracy (near 100%), and precision (CV < 10%) both for intra-day and inter-day experiments were achieved, as well as high recovery rates (>91%). Matrix effects were observed but were compensated by using internal standards. Our newly developed method is applicable for biomonitoring studies as well as large epidemiological studies investigating the effect of oxidative damage, as it requires only minimal clean up using solid phase extraction.

## 1. Introduction

Oxidative stress is a major contributor to the pathophysiology of a variety of diseases [1]. It represents an unbalanced biological state where the natural antioxidant defenses are exceeded due to the presence of reactive oxygen species (ROS). This antioxidant mechanism regulates oxidative stress in the human body against environmental factors such as exposures to UV and pollution, and behavioral habits, such as smoking, diet, drinking, and excessive physical activity as well as ageing and body mass [2]. Excess ROS can cause cellular damage by reacting with cellular components such as proteins, lipids, or DNA [3]. In the human body, oxidative stress plays a crucial role in the onset of several diseases including cancer, diabetes, cardiovascular and respiratory diseases [4,5]. Oxidative stress can both be a cause and a consequence of inflammation. Inflammatory cells such as macrophages and neutrophils are activated upon infection or injury. While fighting off invading pathogens, inflammatory cells produce oxidative stress to an excessive extent, which in turn damages healthy cells, leading to inflammation. Under normal conditions, inflammation decreases after the infection is eliminated or the injury is repaired. Yet, oxidative stress can also trigger the inflammatory response, which generates more oxidative stress, creating a vicious cycle.

ROS concentrations in body fluids cannot be easily quantified as they are highly reactive and have short half-lives. However, biomonitoring of oxidative stress can be achieved by quantifying excreted and stable oxidation products [6]. Several oxidative stress biomarkers in body fluids exist, such as 8-oxo-7,8-dihydro-2′-deoxyguanosine (8-oxodG) and 8-iso-prostaglandin F_2α_ (8-isoprostane). 8-oxodG is one of the major compounds resulting from oxidative damage to DNA [7]. Another name for this biomarker is 8-hydroxy-2′-deoxyguanosine (8-OHdG), a chemically less stable tautomer (Figure 1). The scientific community uses both naming conventions interchangeably [8,9]. The oxidized nucleosides, which are a result of the oxidation of DNA by ROS, are excreted into the urine. Their measurement therefore represents the cumulative total body oxidative stress [10]. In clinical settings, 8-oxodG has been proven to be a predictive factor for the development of diseases. High oxidation of DNA, which is associated with high excretion of urinary 8-oxodG, is predictive for lung and breast cancer risks [11,12].

8-isoprostane is part of the F_2_-isoprostane family. It is formed after oxidation of arachidonic acid, which is present in the membrane phospholipids of the body’s cells [13]. There are 64 F_2_-isoprostane isomers and the most predominant one is 8-isoprostane (also abbreviated as 15-F_2_t-IsoP, 8-iso-PGF_2α_, 8-epi-PGF_2α_, or iPF_2α_-III) (Figure 2) [14]. F_2_-isoprostanes are frequently viewed as the most reliable biomarkers for monitoring oxidative stress in vivo [15,16]. In clinical settings, for example, elevated urinary concentrations of F_2_-isoprostane are found in cardiovascular disease, correlating with severity of disease, and predicting clinical outcomes [17].

Several analytical methods have been developed to quantify 8-oxodG and 8-isoprostane in different biological matrices, including blood, saliva, urine, and exhaled air condensate (EBC). Urine is the preferred matrix in biological monitoring because its collection involves a simple, non-invasive sampling method. Both biomarkers can be quantified by two principal analytical approaches: liquid (LC) or gas (GC) chromatography coupled with mass spectrometry (MS) or enzyme-linked immunosorbent assay (ELISA) [18,19,20]. LC-MS/MS is usually preferred to GC-MS/MS as the latter requires a derivatization step, which introduces possible losses of biomarkers and increases the overall time needed to conduct the analyses. Immunological methods are less sensitive and lack specificity compared to mass-based methods [21]. However, they are still used, as they are faster and do not require expensive analytical instruments.

Four studies have reported concurrent quantification of 8-oxodG and 8-isoprostane in urine by LC-MS/MS. Wu et al. (2016) reported the simultaneous analysis of 8-oxodG, 8-nitroguanine (8-NO_2_Gua), 8-isoprostane, and N-acetyl-S-(tetrahydro-5-hydroxy-2-pentyl-3-furanyl)-L-cysteine (HNE-MA) with solid-phase extraction [22]. Zhao et al. (2017) reported the determination of 8-oxoguanosine, 8-oxodG, and 8-isoprostane with solid-phase extraction [23]. Saito et al. (2018) described the concurrent analysis of 8-isoprostane, 8-oxodG, and 3-nitro-L-tyrosine by online solid-phase microextraction [24]. Martinez and Kannan (2018) reported the determination of 8-oxodG, o-o’-dityrosine (DiY), malondialdehyde (MDA), and four F_2_-isoprostane isomers (including 8-isoprostane) after 2,4-dinitrophenylhydrazine (DNPH) derivatization and solid-phase extraction [25]. Table 1 summarizes the method validation parameters for the different analytical methods. These parameters include limit of detection (LOD), limit of quantification (LOQ), linearity, intra- and inter-day precision and accuracy, recovery, and matrix effects.

Sensitivity of LC-MS methods is dependent on matrix effects when analyzing biological fluids. These effects can be manifested by either a decrease in MS response (signal suppression) or an increase in MS response (signal enhancement) [26]. During method validation, it is important to determine the influence of these matrix effects on MS responses and to find strategies to minimize their impact. It is also advisable to use several sources of biological fluids in this step as matrix effects can vary greatly between urine samples [27]. During our method development, we selected three urine samples with different creatinine concentrations, which represent the different hydration status of the donor. Urine samples with high creatinine concentrations contain more matrix components that can affect the analysis of the biomarkers of interest. We propose several recommendations to reduce or control matrix effects.

This study aimed to optimize the simultaneous analysis of 8-oxodG and 8-isoprostane in urine by LC-MS/MS and to validate a new method following the US Food and Drug Administration (FDA) guidelines for bioanalytical method validation. Our method included the development of a sample preparation procedure (solid-phase extraction) and the optimization of the LC-MS parameters. We applied the method to urine samples of ex-smokers known to have low concentrations of these biomarkers. We confirmed the non-smoking status of the participants by analysis of nicotine and its metabolites in their urine (total nicotine equivalent <2 nmol/mg creatinine). The ranges of creatinine-adjusted 8-oxodG and 8-isoprostane concentrations were in agreement with the reference values reported in the general population. Therefore, non-smokers can be used as controls in oxidative stress research.

## 2. Materials and Methods

### 2.1. Standards, Chemicals, and Material

8-OxodG (≥98% (TLC), CAS Number 88847-89-6) was obtained from Merck KGaA (Buchs, St. Gallen, Switzerland). The isotopically labelled [15N5]-8-oxodG (CAS Number 569649-11-2) was used as the internal standard (IS) and bought from Cambridge Isotope Laboratories, Inc. (Tewksbury, MA, USA). 8-Isoprostane ((5Z,8β,9α,11α,13E,15S)-9,11,15-trihydroxyprosta-5,13-dien-1-oic acid; ≥95%, CAS Number 27415-26-5) and its deuterated isomer (IS) 8-isoprostane-d4 ((5Z,8β,9α,11α,13E,15S)-9,11,15-trihydroxyprosta-5,13-dien-1-oic-3,3,4,4-d4 acid; CAS Number 211105-40-7), were obtained from Cayman Chemical (Ann Arbor, MI, USA). LC-MS grade solvents, water, methanol, and acetonitrile, were obtained from Carlo Erba Reagents (Chaussée du Vexin, Val de Reuil, France). LC-MS grade acetic acid was obtained from Honeywell (Seelze, Germany). MilliQ water was produced in the laboratory with a water purification system (MilliQ Advantage) from Merck (Schaffhausen, Switzerland). Solid phase extraction (SPE) cartridges (Chromabond C18 ec SPE 500 mg 3 mL) were purchased from Macherey-Nagel (Oensingen, Switzerland).

### 2.2. Urine Samples for Method Validation

Three urine samples collected from healthy, consenting adults were aliquoted in tubes (8 mL) before storing at −20 °C. We chose to focus our study on volunteer hydration to investigate its consequences on matrix effects; thus, we chose urine samples by color and creatinine concentration. Indeed, even if urine is a relatively clean matrix, it contains many compounds that can interfere with the analysis [28]. This is especially important with high creatinine urine samples (“dark urine”, indicative of poor hydration). Two samples were chosen to reflect extreme cases: light colored urine or “light urine” with a creatinine concentration of 0.2 mg/mL, and dark colored urine or “dark urine” with a creatinine concentration of 3.58 mg/mL. Both samples were used for method validation. A third urine sample was an intermediate colored urine or “medium urine” with a creatinine concentration of 1.65 mg/mL. The latter urine sample was used to prepare the calibration standards during the method development and validation. Thus, we conducted the method development on three urine samples with different creatinine concentrations.

### 2.3. Calibration Curve and Quality Controls (QC)

Calibration curves were prepared by spiking “medium urine” with six different concentrations of 8-oxodG and 8-isoprostane and a constant concentration of IS (standard stock solution description in Appendix A). The six concentration levels were 0.5, 1, 2.5, 5, 10, and 20 ng/mL for 8-oxodG and 0.1, 0.2, 0.5, 1, 2.5, and 5 ng/mL for 8-isoprostane.

The “light urine” was used to prepare low QC concentrations: 0.63 ng/mL of 8-oxodG and 0.10 ng/mL of 8-isoprostane. The “medium urine” was used to prepare the high QC concentrations: 3.30 ng/mL of 8-OHdG and 0.45 ng/mL of 8-isoprostane.

### 2.4. Solid-Phase Extraction (SPE)

Urine samples were thawed at room temperature and vortexed. Urine amounts for analysis were adjusted according to the creatinine concentration: 500 µL of urine for 1 mg/mL of creatinine (in other words, we adjusted the urine volume to load 0.5 mg of creatinine). Water (400 µL), IS (100 µL), and 10% formic acid (100 µL) were added to form the SPE loading solution. SPE cartridges were first conditioned with methanol (2 mL) and water (2 mL). Urine samples were loaded onto the SPE, washed with water (2 mL) and then 5% methanol (2 mL), dried with air (PRESSURE+ from Biotage, Uppsala, Sweden), and eluted with methanol (3 mL). The extract was filtered (0.45 µm), evaporated under a nitrogen flow with a Pierce Reacti-Therm III evaporator (Thermo Scientific, Reinach, Switzerland Switzerland), and reconstituted in the injection solvent (500 µL 0.1% acetic acid in water). Calibration standards and QC were treated identically to the samples. 

### 2.5. LC-MS/MS Analysis

Analysis of 8-oxodG and 8-isoprostane was performed using a UPLC (Dionex Ultimate 3000 system, Thermo Scientific, Reinach, Switzerland) coupled with a triple-stage quadrupole mass spectrometer (TSQ Quantiva, Thermo Scientific, Reinach, Switzerland) equipped with a heated electrospray ionization source (ESI) operated in positive ion mode for 8-oxodG and in negative ion mode for 8-isoprostane.

The compounds were separated using a C18 column (2.1 × 100 mm, 1.8 µm; Zorbax Eclipse Plus, Agilent, Morges, Switzerland). The column temperature was maintained at 30 °C. The mobile phase consisted of: eluent A composed of 0.1% acetic acid in water, and eluent B of 0.1% acetic acid in methanol/acetonitrile (7:3, v/v). The solvent gradient program was: t = 0 min: 0% B, t = 1.1 min: 55% B, t = 12 min: 65% B, t = 12.5 min: 90% B, t = 14.5 min: 90% B, t = 15.5 min: 0% B, t = 22 min: 0% B, at a flow rate of 250 µL/min. Using methanol and acetonitrile mixture as the mobile phase (B) was based on previous work from Prasain et al. (2013) reporting that F_2_-isoprostane isomers’ separation was not achieved with a mobile phase of 100% methanol [29].

Multi-reaction monitoring (MRM) transitions and ESI parameters can be found in Appendix A. All data acquisition and processing were accomplished using the Thermo Scientific Chromeleon software (version 7.2.10).

### 2.6. Application to Urine Samples Obtained from Healthy Participants

For method application, urine samples were collected from an on-going randomized controlled trial on smoking cessation: “Efficacy, Safety, and Toxicology of Electronic Nicotine Delivery Systems as an aid for smoking cessation: the ESTxENDS multicenter randomized controlled trial” (ClinicalTrials.gov Identifiers: NCT03589989) approved by the Ethics committees of Bern, Geneva, and Lausanne (Project-ID: 2017-02332), Switzerland. The study was conducted in accordance with the ethical principles of the World Medical Association Declaration of Helsinki and the International Committee on Harmonization for Good Clinical Practice and Swiss law. All participants provided a written informed consent and the following information: age, gender, and anthropometric data (height, weight).

For this study, we selected participants who reported that they were cigarette abstinent for more than four months, were not using any other nicotine delivery systems (e-cigarettes, nicotine replacement therapy or any other nicotine containing device) and had donated their first-void urine sample (first morning urine sample). We validated the smoking abstinence by assessing total urinary nicotine equivalent (<2 nmol/mg creatinine). The urine samples were first stored at 4 °C (for 1 to 7 days), and urine aliquots were then stored at −20 °C until analysis.

### 2.7. Participant Description

Fifty-six participants provided first-void urine samples for the quantification of 8-oxodG and 8-isoprostane. Mean age of the participants was 43.5 years old with a BMI mean of 26. Twenty-six participants were women (46%) and 30 participants were men (54%). Participant demographics are described in Table 2.

### 2.8. Other Bioanalytical Methods

Urinary concentrations of 8-oxodG and 8-isoprostane were adjusted with creatinine concentration to account for the hydration status of the participants and allow inter-individual comparison. There is an acceptable correlation between creatinine corrected spot-urine and 24 h urine [30,31,32,33]. Creatinine was quantified at the Unit of Forensic Toxicology and Chemistry, University Center of Legal Medicine (Lausanne—Geneva, Switzerland) with a routine clinical method based on Jaffe (1886) [34].

Total nicotine equivalent (TNE) is considered as the gold standard biomarker of daily nicotine intake [35]. In most studies, TNE is based on six metabolites (nicotine, cotinine, trans-3′-hydroxycotinine, cotinine-*N*-glucoronide, nicotine-*N*-glucoronide, and trans-3-hydroxycotinine-*O*-glucoronide). In this study, only four metabolites were included (TNE 4) as it was sufficient for smoking status verification. Nicotine, cotinine, trans-3′-hydroxycotinine, and norcotinine were analyzed at the Unit of Forensic Toxicology and Chemistry, University Center of Legal Medicine (Lausanne—Geneva, Switzerland) by LC-MS/MS with a routine method based on an application note of Thermo Fisher Scientific (n°20709, 2013). TNE was calculated as TNE = (nicotine/162.23 + cotinine/176.22 + trans-3′-hydroxycotinine/192.22 + norcotinine/162.19)/creatinine, expressed in nmol/mg creatinine).

### 2.9. Data Presentation and Statistical Analysis

Method validation parameters were calculated based on the peak areas that were integrated by the UPLC-MS/MS software. Description of these parameters can be found in Appendix A. Total nicotine equivalent was calculated as the molar sum of nicotine, cotinine, trans-3′-hydroxycotinine, and norcotinine (corrected by the creatinine concentration). Oxidative stress biomarkers were creatinine-corrected and were presented as median with the 1st and 3rd quartile. All calculations were performed with the R program (R version 3.6.2 (12 December 2019)—“Dark and Stormy Night”).

## 3. Results

### 3.1. LC-MS/MS Analysis

After the SPE on C18 cartridge (optimization description in Appendix A), the samples were analyzed by LC-MS. LC separation was performed on a C18 column with a gradient of 0.1% acetic acid in water (A) and 0.1% acetic acid in methanol/acetonitrile 70:30 (%, *v*/*v*; B) at a flow rate of 250 µL/min. Retention times were 4.7 min for 8-oxodG and 10.2 min for 8-isoprostane (Figure 3). Internal standards’ retention times were similar. Elution of 8-oxodG and its internal standard occurred after the solvent front, indicating that the column did retain the compound. This also helped to reduce the signal suppression during the mass spectrometry process.

ESI mode interface was operated in positive ion mode for the first segment of the run (0.5–8 min) and in negative ion mode for the second segment (8.5–14 min) to optimize the detection of both analytes. Ion source parameters, as well as m/z transitions for the multiple reaction monitoring, were determined by infusion of aqueous standard of 8-oxodG (5 µg/mL) and 8-isoprostane (5 µg/mL). Mass transitions, collision energy, and RF lens are shown in Appendix A (Appendix A).

MRM transitions for 8-isoprostane showed the probable presence of other F_2_-isoprostane isomers in urine (Figure 4). Separation gradient was optimized to allow the peak separation of 8-isoprostane with other potential isomers in urine samples.

We tested a lower concentration of acetic acid in the mobile phase (0.01%), but it did not increase 8-isoprostane signal and decreased 8-oxodG signal. Use of formic acid (0.1%) reduced the signals of both analytes.

### 3.2. Sensitivity, Linearity, Accuracy, and Precision

We determined LODs at 10 pg/mL for 8-oxodG and 20 pg/mL for 8-isoprostane (S/N ≥ 3) and the LOQs at 30 pg/mL for 8-oxodG and 50 pg/mL for 8-isoprostane (S/N ≥ 10 and coefficient of variation <20%) in aqueous solution. In urine, our lowest calibration standard was 0.5 ng/mL for 8-oxodG and 0.1 ng/mL for 8-isoprostane. These concentrations were low enough to quantify these biomarkers in participants’ urine samples and were in accordance with previous published methods. Therefore, calibration curves were constructed with six levels from 0.5 to 20 ng/mL for 8-oxodG and 0.1 to 5 ng/mL for 8-isoprostane in urine. Linear regression with 1/x weighting was performed on analyte/IS peak area ratio versus standard concentrations. Linearity of the working ranges was observed with a regression coefficient of R^2^ > 0.999. Slopes of the calibration curves were similar for urine and water: 2% ± 7% for 8-oxodG and 3% ± 6% for 8-isoprostane.

Intra-day precision and accuracy were determined by analyzing three replicates of two urine samples spiked at three concentrations: 0.5, 1, and 10 ng/mL for 8-oxodG and 0.1, 0.2, and 0.5 ng/mL for 8-isoprostane. Intra-day accuracy ranged from 92% to 114% with a coefficient of variation lower than 5.7% for 8-oxodG, and from 97% to 114% with a coefficient of variation lower than 7% for 8-isoprostane. Injections were performed for three days to determine the inter-day precision and accuracy. The inter-day accuracy for 8-oxodG ranged from 92% to 103% with a coefficient of variation lower than 10%, and from 97% to 114% with a coefficient of variation lower than 8.1% for 8-isoprostane. Accuracy and precision details are shown in Table 3.

### 3.3. Extraction Recovery and Matrix Effects

During the method development, extraction recoveries were calculated for each concentration used in the calibration curve. We observed stable extraction recoveries. The extraction recovery and the matrix effects for the concentrations corresponding to the highest calibration curve levels, 20 ng/mL for 8-oxodG and 5 ng/mL for 8-isoprostane, are presented. To determine extraction recovery, three replicates in urine spiked before SPE with 8-oxodG and 8-isoprostane and three replicates in urine spiked after SPE with the same solution were analyzed. Extraction recovery was 97% for 8-oxodG and 91% for 8-isoprostane. To determine absolute matrix effects, three replicates in water spiked with 8-oxodG and 8-isoprostane (without SPE) were compared to three replicates in urine spiked after SPE with the same solution. Matrix effects were found to be urine-dependent, and we observed matrix effects up to 20% for 8-oxodG and 70% for 8-isoprostane for “medium urine” (100% corresponds to no matrix effects). Variation of the analyte to IS ratio was lower than 4% indicating that the observed signal reduction was compensated by using a stable isotopic internal standard.

Matrix effects were also observed for “light urine” (67% for 8-oxo-dG and 83% for 8-isoprostane) and “dark urine” (4% for 8-oxodG and 25% for 8-isoprostane), estimated by the IS variation. A simple dilution by a factor of two of the “dark urine” reduced matrix effects to 19% for 8-oxodG and 58% for 8-isoprostane (more information in the Appendix A). This indicated that signal suppression can be reduced by diluting urine samples prior to analysis or by taking a lower volume of urine for analysis.

Relative matrix effects were estimated by comparing the slopes of calibration curves in three different urine samples. Coefficient of variation for the slopes of both 8-oxodG and 8-isoprostane were <5%, which emphasizes the importance of the stable isotopic internal standard for matrix effect correction.

### 3.4. Stability

We evaluated stability of 8-oxodG and 8-isoprostane in urine at −20 °C by monitoring the QC (low and high) over a 6-month period. 8-oxodG concentration was 8–9% higher after 6 months for both low (0.65 ng/mL) and high QC (3.42 ng/mL). The variation of the concentration over the whole period (65 injections) was less than 5% for both. 8-isoprostane concentration was 15% lower after 6 months for low QC (0.09 ng/mL) and 7% higher for high QC (0.46 ng/mL). The variation of the concentration over the whole period (65 injections) was 13% and 7% for low and high QC, respectively.

Stability of the analytes in processed urine at room temperature was also monitored by the QC (low and high). QCs were injected three times in an injection sequence (at the beginning, in the middle, and at the end), seven hours apart. The average of the intra-sequence variation of 8-oxodG was 1.42% and 1.34% for low and high QC, respectively. The average of the intra-sequence variation of 8-isoprostane was 8.9% and 6.1% for low and high QC, respectively. There was no tendency for signals to increase or decrease between the 1st and the 3rd injection (i.e., after about 14 h), meaning that the analytes were stable in processed urine during this period.

### 3.5. Oxidative Stress Biomarkers’ Concentrations in Healthy Participants

Oxidative stress biomarkers’ concentrations were determined in 56 morning urine samples obtained from the participants. The two analytes were quantified in all samples. After creatinine correction, the median of 8-oxodG concentration was 4.04 ng/mg creatinine (1st quartile–3rd quartile: 3.42–5.37 ng/mg creatinine) and the median of 8-isoprostane concentration was 0.23 ng/mg creatinine (1st quartile–3rd quartile: 0.14–0.28 ng/mg creatinine). Details are shown in Table 4.

## 4. Discussion

We successfully optimized the simultaneous quantification of urinary 8-oxodG and 8-isoprostane by LC-MS/MS and efficiently applied the method to 56 urine samples from participants (non-smoking status confirmed by total nicotine equivalent <2 nmol/mg creatinine). The creatinine-adjusted concentrations ranges of 8-oxodG and 8-isoprostane were in agreement with the reference values of the population [36,37]. This is interesting, because this would allow non-smokers to be used as controls in studies investigating the effects of a particular exposure (e.g., air pollution, UV) or behavioral habit (e.g., smoking, intense activity) on the oxidative stress level.

Matrix effects are commonly observed in analysis of biological fluids and can obscure the signal in an otherwise selective and sensitive LC-MS method. The matrix effects’ mechanisms are not fully understood, but they involve co-elution of matrix components that induce a loss of response (signal suppression) or an increase of response (signal enhancement). As all urine samples have different compositions, Matuzewski et al. (2003) recommended performing method validation in five different sources instead of a single one [27]. 

We hypothesized that “dark urine” samples (from individuals with a low hydration status) cause greater matrix effects than “light urine” samples (from individuals with a high hydration status). We selected urine samples according to the aspect (color) and urinary creatinine concentration. The latter is dependent on hydration status, as an increased amount of water in urine will lower the creatinine concentration. We demonstrated that matrix effects were proportional to urinary creatinine concentrations. This finding is of primary importance, because even if the matrix effect is compensated by the use of internal standards, the sensitivity of the method is decreased due to signal suppression (comments on matrix effects and method performance in Appendix A).

Therefore, it is highly recommended to construct calibration curves in the same biological fluid as the samples. It is also important to assess relative matrix effects by comparing calibration curve slopes constructed in different urine samples. Similar slopes indicate that sample matrix and recovery differences do not alter precision and accuracy. This is an additional argument for the use of multiple urine sources during method development and validation. In order to have comparable MS response intensities for urine samples, we adjusted the urine volume according to the creatinine concentration.

As oxidative stress biomarkers are usually corrected by creatinine concentrations for spot urine samples, it is therefore reasonable to use these known concentrations during sample preparation. It would also be possible to adjust the volume of urine used for analysis by the density or the total urine volume. We chose 500 µL of urine for 1 mg/mL creatinine because 1 mg/mL is close to the average creatinine concentration in spot urine samples (1.3 mg/mL [38]). Furthermore, we obtained good precision and accuracy for low concentrations (0.5 ng/mL for 8-oxodG and 0.1 ng/mL for 8-isoprostane). This allows us to control for matrix effects.

We planned initially to include malondialdehyde (MDA), with 8-oxodG and 8-isoprostane, in the method. Simultaneous analysis of different biomarkers presents many advantages such as saving time and money. It can be challenging if the analytes have different physicochemical properties. This is the case for 8-oxodG, 8-isoprostane, and MDA. 8-oxodG is composed of a purine and is a polar molecule due to the presence of polar functional groups (amides, hydroxyls, and amine). Moreover, it is uncharged under low and neutral pH and forms anions and dianions at higher pH values (pKa values at 8.6 and 11.7) [39,40]. 8-isoprostane is mostly non-polar due to its alkane chains. Nevertheless, solubility in water is possible due to the presence of polar functional groups (hydroxyls and carboxyl). The molecule is neutral in acidic conditions and forms an anion under neutral and alkaline conditions (pK_a_ value at pH ~5) [41]. MDA is a small, reactive molecule that undergoes keto-enol tautomerism. Most of the analytical methods involve a derivatization step [42].

We used dinitrophenylhydrazine (DNPH) solution (5 mM) in water/acetonitrile/acetic acid (6.5:1:2.5) for the derivatization as described by Martinez and Kannan (2018) [25]. However, we were not able to reproduce the results because 8-oxodG was not completely retained on the SPE cartridge during the loading and washing steps. This could be due to the presence of acetonitrile from the MDA derivatization step prior to SPE. We performed the tests with the four non-polar cartridges listed in Appendix A (Appendix A) without success. We therefore decided to exclude MDA from the method to improve the retention of 8-oxodG. We decided to keep 8-oxodG as it is an important biomarker of DNA damage, and not include MDA, as MDA and 8-isoprostane are both biomarkers of lipid peroxidation. Moreover, the physiological validity of MDA as a biomarker of oxidative stress is rather poor. The main reasons are that MDA formation is not specific to lipid peroxidation, there is a lack of association between MDA and oxidative stress in humans, and urinary MDA concentrations are potentially also modified by diet [43].

We were able to quantify low concentrations of oxidative stress biomarkers (0.5 ng/mL for 8-oxodG and 0.1 ng/mL for 8-isoprostane) in 56 morning urine samples from non-smoking healthy participants. Matrix effects were observed during the analysis of urine samples and their magnitude was directly linked to the urinary creatinine concentration, a measure of hydration level of the individual. This also meant that the sample clean-up was not complete as matrix components induce MS signal suppression effects. Solid-phase extraction with a reversed-phase cartridge was chosen because it retained both analytes well. However, 8-oxodG is more polar than 8-isoprostane and eluted with low percentage of methanol (10%). Therefore, the SPE washing step could not be optimized further to remove more matrix components without losing 8-oxodG. Both 8-oxodG and 8-isoprostane are negatively charged at high pH. Nevertheless, 8-isoprostane was not recovered during the elution step for the two anionic SPE cartridges we tested. Due to the different physicochemical properties of the two analytes, the reversed-phase was a good compromise.

The various existing F_2_-isoprostane isomers can complicate the 8-isoprostane quantification due to possible co-elution [29]. As we observed several peaks on the UPLC chromatograms close to the retention time of 8-isoprostane, we adjusted the separation gradient to isolate the 8-isoprostane peak. This was the main reason why the total duration of the analytical run could not be shortened. Other peaks might be other F_2_-isoprostane isomers but this was not explored further.

The obtained concentration medians for 8-oxodG (4.04 ng/mg creatinine) and 8-isoprostane (0.23 ng/mg creatinine) in our participants were comparable to the values in healthy adults reported in two systematic reviews by Graille et al. (2020a, 2020b) [36,37]. For 8-oxodG, the authors reported a median value obtained by chromatographic analytical techniques in healthy adults of 3.9 ng/mg creatinine (3–5.5 ng/mg creatinine) with a BMI ≤ 25 and 2.8 ng/mg creatinine (2.4–3.5 ng/mg creatinine) with a BMI > 25. For 8-isoprostane, they reported a median value of 0.249 ng/mg creatinine (0.186–0.407 ng/mg creatinine) for healthy adults with a BMI ≤ 25 and 0.508 ng/mg creatinine (0.180–0.553 ng/mg creatinine) for healthy adults with a BMI > 25. Therefore, our method is sufficiently sensitive in quantifying background population concentrations of these oxidative stress biomarkers and could be applied in clinical or epidemiological studies.

Oxidative stress biomarkers and inflammation markers have been analyzed together in several studies. Helmersson et al. (2004) and Tatsch et al. (2015) showed respectively that type 2 diabetes led to chronic inflammation followed by oxidative damage and that patients with higher 8-oxodG concentrations had higher degrees of inflammation and higher insulin resistance [44,45]. Altemose et al. (2017) and Squillacioti et al. (2020) reported that exposure to air pollution (including PAHs and aldehydes) contributed to the induction of oxidative stress and airways inflammation [46,47]. Ochoa et al. (2011), Mrakic-Sposta et al. (2015), and Larsen et al. (2020) investigated the effect of intense exercise on elevation of oxidative stress and inflammation markers [48,49,50]. Several researches have been conducted on the effect of diet on these markers, including those of Helmersson et al. (2008) and Holt et al. (2009), which showed the importance of a healthy diet and highlighted the beneficial effect of fruits and vegetables [51,52].

Of these nine studies, only one used LC-MS as an analytical technique and only two studies quantified both analytes with two separate analyses. The method we propose would provide a simultaneous quantification of 8-oxodG and 8-isoprostane. By addressing both analytes in one run, this method saves time and consequently money and can thus be used in larger epidemiological studies. This method can help in gaining a better understanding of the relationship between oxidative stress and inflammation, and to understand the underlying mechanisms as currently these biomarkers are not completely understood. We would also highlight that the measurement variability of our method is lower than the intra-individual variabilities for both 8-oxodG and 8-isoprostane, which renders them excellent in molecular epidemiological studies [30,53].

From a clinical perspective, 8-isoprostane and 8-oxodG are important oxidative stress biomarkers, which have a diagnostic and prognostic value and correlate with disease degree. Therefore, screening for 8-isoprostane and 8-oxodG in a fast and cost-effective way could help to identify people at risk and monitor the potential effect of interventions.

## 5. Conclusions

Our concurrent analysis of urinary 8-oxodG and 8-isoprostane method is rapid, stable, and robust. We recommend using a stable isotopic internal standard to compensate for matrix effects. The matrix effect was related to creatinine content; consequently, we suggest diluting “dark urine” (high creatinine concentration) to reduce ion suppression effects and increase the loading volume of “light urine” (low creatinine concentration) to allow quantification. We successfully analyzed 56 urine samples from healthy non-smoking participants and were able to quantify background levels of oxidative stress biomarkers. Our method is suitable for large epidemiological or biomonitoring studies.

## Figures and Tables

**Figure 1 antioxidants-10-00038-f001:**
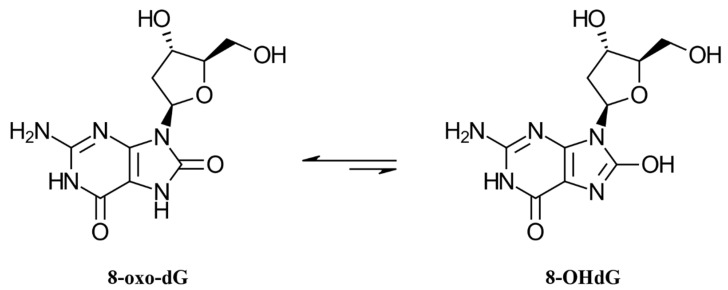
Structure of 8-oxo-7,8-dihydro-2′-deoxyguanosine (8-oxodG) and its tautomer 8-OHdG.

**Figure 2 antioxidants-10-00038-f002:**
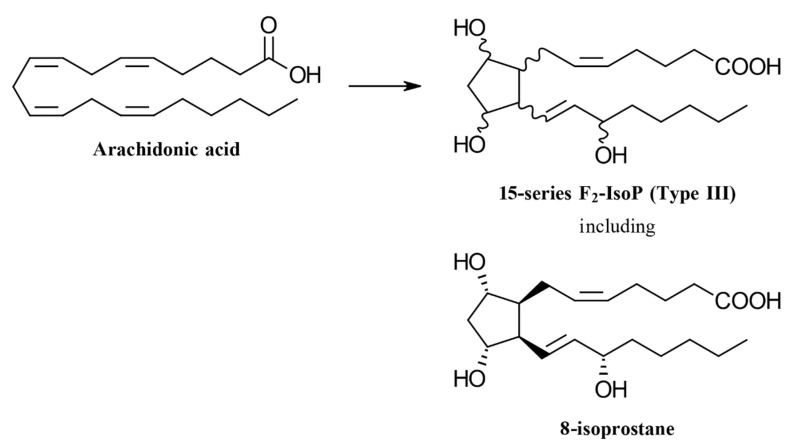
8-Isoprostane is one of the 64 isomers formed by the oxidation of arachidonic acid. It is part of the 15-series F_2_-isoprostanes.

**Figure 3 antioxidants-10-00038-f003:**
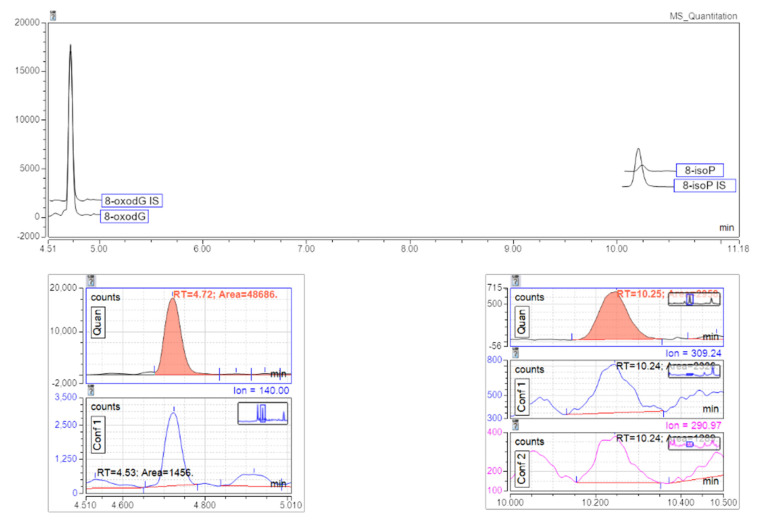
Chromatogram of spiked “light urine”; chromatogram with retention times and multi-reaction monitoring (MRM) transitions for 8-oxodG (left) and 8-isoprostane (right).

**Figure 4 antioxidants-10-00038-f004:**
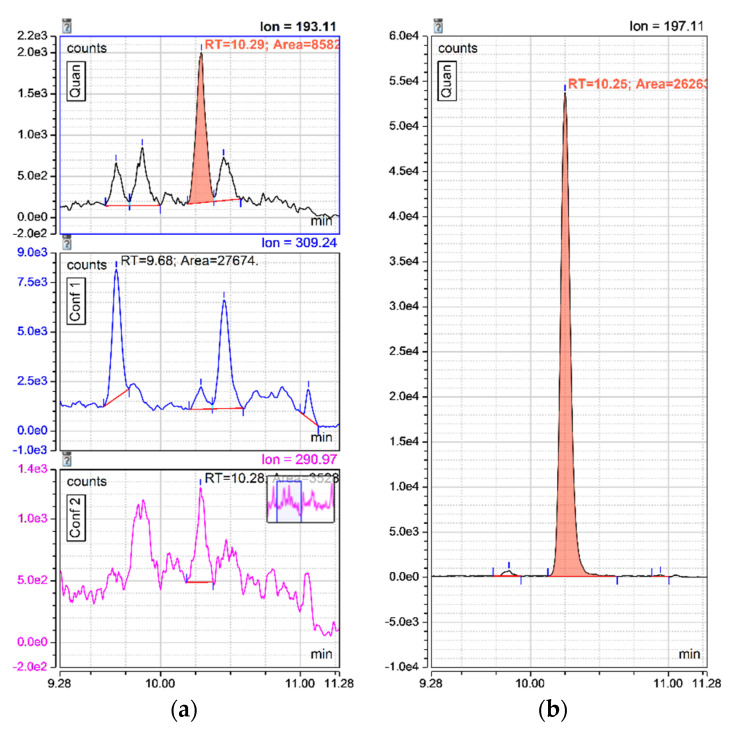
(**a**) MRM transitions for 8-isoprostane in “light urine”; presence of potential other F_2_-isoprostane isomers; (**b**) corresponding 8-isoprostane-d_4_ chromatogram.

**Table 1 antioxidants-10-00038-t001:** Descriptions of the multi-analyte analytical methods previously developed and our method.

	Wu et al. (2016) [22]	Zhao et al. (2017) [23]	Saito et al. (2018) [24]	Martinez and Kannan (2018) [25]	Our Study
**Parameters**					
**LOD**					
8-oxodG	20 pg/mL	170 pg/mL	12.6 pg/mL	30 pg/mL	10 pg/mL
8-isoprostane	8 pg/mL	40 pg/mL	3.4 pg/mL	10 pg/mL	20 pg/mL
**LOQ**					
8-oxodG	50 pg/mL	570 pg/mL	20 pg/mL	100 pg/mL	30 pg/mL
8-isoprostane	30 pg/mL	130 pg/mL	29 pg/mL	20 pg/mL	50 pg/mL
**Linearity**					
8-oxodG	R^2^ > 0.998	R^2^ > 0.999	R^2^ > 0.999	R^2^ > 0.999	R^2^ > 0.999
8-isoprostane	R^2^ > 0.998	R^2^ > 0.999	R^2^ > 0.999	R^2^ > 0.999	R^2^ > 0.999
**Intra-/inter-day accuracy**					
8-oxodG 8-isoprostane	98.8–102.2%/	n.a./	91.1–97%/	92–101%/	92–114%/
	98.5–101.6%	n.a.	n.a.	n.a.	92–103%
	98.5–101.7%/	n.a./	95.7–100%/	93–103%/	97–114%/
	99–102.1%	n.a.	n.a.	n.a.	97–114%
**Intra-/inter-day precision**					
8-oxodG	<8.1%/<8.5%	<1.9%/<3.9%	<5%/<6.1%	<9%/n.a.	<5.7%/<10%
8-isoprostane	<4.6%/<5.1%	<2.3%/<5.3%	<2.1%/<4.5%	<9%/n.a.	<7.0%/<8.1%
**SPE recovery**					
8-oxodG	90.1–90.7%	90.1–100%	n.a.	n.a.	97%
8-isoprostane	94.3–95%	89.2–108%	n.a.	n.a.	91%
**Matrix effects** ^1^					
8-oxodG	89.2%	n.a.	n.a.	n.a.	20%
8-isoprostane	96.6%	n.a.	n.a.	n.a.	70%

^1^ Absolute matrix effects.

**Table 2 antioxidants-10-00038-t002:** Summary of participant demographics and verification of smoking abstinence.

Characteristic	Non-Smokers (*n* = 56)
Age (years)	43.5 (35.5–54.25) *
Sex	
Men	30 (54) **
Women	26 (46)
BMI (kg/m^2^)	26 (23–28)
≤25	22 (39)
>25	34 (61)
TNE (nmol/mg creatinine)	0.01 (0.00–0.02)

* Median (IQR: 25–75%); ** Number (% of total).

**Table 3 antioxidants-10-00038-t003:** Accuracy and precision for 8-oxodG and 8-isoprostane at three concentrations in two different urine samples.

Urine	Compound	Concentration ^1^	Intra-Day Accuracy ^2^ and Precision ^3^		Inter-Day Accuracy and Precision	
“Light urine”	8-oxodG	0.5	94	0.9	94	2.5
		1	92	1.9	92	2.7
		10	99	0.5	99	2.4
	8-isoprostane	0.1	97	5.9	97	8.1
		0.2	99	4.4	99	3.4
		0.5	100	3.3	100	2.7
“Dark urine”	8-oxodG	0.5	113	5.4	103	10
		1	107	5.7	100	8.4
		10	98	0.8	99	4.4
	8-isoprostane	0.1	114	7.0	114	2.1
		0.2	100	5.5	100	4.9
		0.5	102	2.0	102	2.5

^1^ [ng/mL]; ^2^ %; ^3^ coefficient of variation [%].

**Table 4 antioxidants-10-00038-t004:** 8-oxodG and 8-isoprostane concentrations in participants’ urine.

8-oxodG			8-isoprostane		
All Participants	BMI		All Participants	BMI	
4.04 * (3.42–5.37)	BMI ≤ 25 (*n* = 22)	4.28 (3.62–6.11)	0.20 (0.14–0.28)	BMI ≤ 25 (*n* = 22)	0.19 (0.14–0.30)
	BMI > 25 (*n* = 34)	3.96 (2.81–4.97)		BMI > 25 (*n* = 34)	0.21 (0.14–0.27)

* Median (IQR: 25–75%), expressed in ng/mg creatinine.

## Data Availability

The data presented in this study are available on request from the corresponding author. The data are not publicly available because the ESTxENDS trial is ongoing. Additional data are provided in Appendix A.

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
