# Peer review of "Rapid Liquid Chromatography—Tandem Mass Spectrometry Analysis of Two Urinary Oxidative Stress Biomarkers: 8-oxodG and 8-isoprostane"

_antioxidants, 2020, doi:10.3390/antiox10010038_

Round 1
Reviewer 1 Report
I would suggest that authors send the manuscript to a journal of analytical chemistry or medicinal chemistry.
Author Response
Comments Reviewer 1
Point 1: I would suggest that authors send the manuscript to a journal of analytical chemistry or medicinal chemistry.
Response point 1:
We thank Reviewer 1 for his comment. We moved some analytical method details to supplementary information and added paragraphs on the usefulness of the method in the context of oxidative stress research. New paragraphs begin at line 404 and 415 (see below).
“Oxidative stress biomarkers and inflammation markers have been analyzed together in several studies. Helmersson et al. (2004) and Tatsch et al. (2015) showed respectively that type 2 diabetes leaded to chronic inflammation followed by oxidative damages and that patients with higher 8-oxodG concentrations had higher degree of inflammation and higher insulin resistance [44,45]. Altemose et al. (2017) and Squillacioti et al. (2020) reported that exposure to air pollution (including PAHs and aldehydes) contributed to the induction of oxidative stress and airways inflammation [46,47]. Ochoa et al. (2011), Mrakic-Sposta (2015), and Larsen et al. (2020) investigated the effect of intense exercise on elevation of oxidative stress and inflammation markers [48–50]. Several researches have been conducted on the effect of diet on these markers, including those of Helmersson et al. (2008) and Holt et al. (2009), which showed the importance of a healthy diet and highlighted the beneficial effect of fruit and vegetables [51,52].”
“Of these nine studies, only one used LCMS as an analytical technique and only two studies quantified both analytes, with two separate analyses. The method we propose would provide a simultaneous quantification of 8-oxodG and 8-isoprostane. By addressing both analytes in one run, this method saves both time and consequently money and can thus be used in larger epidemiological studies. This method can help in gaining a better understanding of the relationship between oxidative stress and inflammation and to understand the underlying mechanisms as currently these biomarkers are not completely understood. We would also highlight that the measurements variability of our method is lower than the intra-individual variabilities for both 8-oxodG and 8-isoprostane, which renders them excellent in molecular epidemiological studies [30,53].”
Reviewer 2 Report
The submitted paper "Rapid liquid chromatography – tandem mass
3 spectrometry analysis of two urinary oxidative stress
4 biomarkers: 8-oxodG and 8-isoprostane" presents interesting dataon concurrent analysis of two biomarkers of oxidative stress.
The selection and representation of biomarkers is well explained and the presentation of the method is quite extensive.
The experimental part is detailed and the description of results may seem ovewhelming. The impact of the amount of analytical data and parameters makes this part difficult to read due to the amount of details and complexity of the whole procedure. However, transferring parts of the text to supplementary data may not work so well, on the other hand, a bit longer introduction to planned research (for example, creatinine reference concentration and nicotine study).
I think the experience derived from preparing the two metadata papers (ref. 36 and 37) positevely affected the perception of detail of the Authors for the mixed (unfortunately) benefit of the readers.
The description of matrix effects and the fragment from line 328 may cause problems for readers not familiar with ESI-MS operation. (incerease in signal intensity after dilution, different for two analytes).
In their study Authors did not discuss the effect of solvent composition (from LC) on MS signal intensity. The difference in retention time makes the amount of organic phase entering the ion source a substantial factor in sensitivity considerations. The mobile phase was acidified with acetic acid, which may have an effect on 8-isoprostane detection levels in negative ion mode. The mentioned application of formic acid (lines 288-289) reduced signal of both analytes - but did it affect retention times?
The presentation of mobile phase composition in gradient mode as %A is less common than using %B.
The gradient was optimized, however, the reason for using methanol-acetonitrile phase was not mentioned (?).
The anion exchange cartridges in SPE did not work well for 8-isoprostane. As the urine sample was acidified before SPE, did the Authors consider pH effect on SPE efficiency?
The retention time in LC, retention on SPE anion exchanger and MS ionization may sugget that it is worth extending the statement:
"8-oxodG is composed of a purine and is a polar molecule due to the presence of polar functional groups (amides, hydroxyls, and amine). Moreover, it is uncharged under neutral pH and forms anions and dianions at higher pH values (pKa values at 8.6 and 11.7)"
to lower pH situation.
The chemical analytical techniques (line 453) could be specified as this phrase brings to mind a wet-technique analysis and derivatisation. (physicochemical, as mostly HPLC and MS are used?)
In conclusion, I find this text really interesting and full of useful information, however, the details require careful reading to understand the significance of experimental design.
Comments on the text preparation:
There are insonsitencies between the text and data in table 3.
In the sentence there seem to be a redundance:
(192-193) Criteria for linearity was a coefficient of determination for urinary calibration curve greater than R2>0.999.
There are fragments of the text that may benefit from re-wording:
(52) 8-oxodG is one of the major lesions resulting from oxidative damage to DNA
(72-73) Urine is the preferred matrix in biological monitoring because it is easily obtainable and a non invasive sampling method.
plural forms in several fragments of the text seem strange:
(211) urines samples were
Author Response
Comments Reviewer 2
Point 2: The submitted paper "Rapid liquid chromatography – tandem mass spectrometry analysis of two urinary oxidative stress biomarkers: 8-oxodG and 8-isoprostane" presents interesting data on concurrent analysis of two biomarkers of oxidative stress. The selection and representation of biomarkers is well explained and the presentation of the method is quite extensive. The experimental part is detailed and the description of results may seem overwhelming. The impact of the amount of analytical data and parameters makes this part difficult to read due to the amount of details and complexity of the whole procedure. However, transferring parts of the text to supplementary data may not work so well, on the other hand, a bit longer introduction to planned research (for example, creatinine reference concentration and nicotine study).
Response point 2:
Following the suggestion of Reviewer 2, we added a description of the study design in the introduction. We modified the paragraph at line 95 and 105 (see below).
“During our method development, we selected three urine samples with different creatinine concentrations, which represent different hydration status of the donor. Urine samples with high creatinine concentrations contain more matrix components that can affect the analysis of the biomarkers of interest. We proposed several recommendations to reduce or control matrix effects.”
“This study aimed to optimize the simultaneous analysis of 8-oxodG and 8-isoprostane in urine by LC-MS/MS and to validate a new method following the US Food and Drug Administration (FDA) guidelines for bioanalytical method validation. Our method included the development of a sample preparation procedure (solid-phase extraction) and the optimization of the LCMS parameters. We applied the method to urine samples of ex-smokers known to have low concentrations of these biomarkers. We confirmed the non-smoking status of the participants by analysis of nicotine and its metabolites in their urine (total nicotine equivalent <2 nmol/mg creatinine). The ranges of creatinine-adjusted 8-oxodG and 8-isoprostane concentrations were in agreement with the reference values reported in the general population. Therefore, non-smokers can be used as controls in oxidative stress research.“
Point 3: I think the experience derived from preparing the two metadata papers (ref. 36 and 37) positively affected the perception of detail of the Authors for the mixed (unfortunately) benefit of the readers.
Response point 3:
To facilitate the reading, we reduced the importance of the details in the main text and moved them to the Supplementary Information.
Point 4: The description of matrix effects and the fragment from line 328 may cause problems for readers not familiar with ESI-MS operation. (Increase in signal intensity after dilution, different for two analytes).
Response point 4:
We agree with the reviewer and moved these observations into the Supplementary Information. We added a description of the matrix effects to the Supplementary Information (see below).
“We observed signal suppression due to matrix effects. This signal suppression was proportional to the concentration of the urine sample (i.e. the presence of co-eluting matrix components). This relationship was, however, not linear. This explains why we observed a signal increase after sample dilution. For example, the matrix effect for 8-oxodG (“dark urine”) changed from 4% to 19% with a two-fold dilution. If we assume that the MS signal of the undiluted sample was 1, then the theoretical signal would be 25 (corresponding to 100%; no matrix effect). Diluting by two (and considering a linear response of the instrument with a slope of 1) then the theoretical signal of the two-fold diluted sample would be 12.5 (100%). Applying the matrix effect of 19% would give a signal of 2.4, which is effectively higher than the signal of the undiluted sample (1).
Signal suppression was different between the two analytes. There are more compounds co-eluting with 8-oxodG (4.7 min) than with 8-isoprostane (10.2 min). Generally, the closer the compounds are to the solvent elution (short retention times), the stronger the matrix effects. To counter balance this effect, appropriate internal standard are used for correcting for signal suppression. Stable isotopically labeled internal standards are preferred since their retention times are very close to those of the analytes and will undergo similar matrix effects as the biomarker.”
Point 5: In their study, Authors did not discuss the effect of solvent composition (from LC) on MS signal intensity. The difference in retention time makes the amount of organic phase entering the ion source a substantial factor in sensitivity considerations. The mobile phase was acidified with acetic acid, which may have an effect on 8-isoprostane detection levels in negative ion mode. The mentioned application of formic acid (lines 288-289) reduced signal of both analytes - but did it affect retention times?
Response point 5:
The solvent composition influenced strongly the retention and signal intensity of both analytes. For example, we observed that the retention time for 8-oxodG in the LC column was very sensitive to the organic content of the elution solvent. As soon as some organic solvent was added to the water eluent, the retention time of this analyte was decreased and it co-eluted with matrix compounds, with a concomitant signal intensity reduction. This is the reason why we started the elution with 100% water. For 8-isoprostane and based on data from Prasain et al. (2013), we selected a mixture of MeOH/ACN in order to:
- increase the separation between the different F2-isoprostane isomers. This separation is better when ACN is used;
- increase the signal to noise ratio. For that purpose, MeOH is reported to be the best solvent.
We optimized these parameters, and found that a mixture of MeOH/ACN 70:30 was the best compromise to have specific and sensitive detection of 8-isoprostane. We were aware that working with acidified mobile phase might have a negative effect on 8-isoprostane signal. However, we tested a lower acetic acid concentration (0.01%) and did not observe any signal increase (8-oxodG decreased though; Line 245). That is why we added 0.1% acetic acid in the mobile phase. The replacement of acetic acid by formic acid did not significantly change the retention times of the two compounds. No modifications have been made in the manuscript.
Point 6: The presentation of mobile phase composition in gradient mode as %A is less common than using %B. The gradient was optimized, however, the reason for using methanol-acetonitrile phase was not mentioned (?).
Response point 6:
We modified the solvent gradient program description following the reviewer’s suggestion (line 171):
“The solvent gradient program was: t=0 min: 0% B, t=1.1 min: 55% B, t=12 min: 65% B, t=12.5 min: 90% B, t=14.5 min: 90% B, t=15.5 min: 0% B, t=22 min: 0% B at a flow rate of 250 µL/min.”
We adapted the solvent mixture used for the mobile phase from the method published by Prasain et al. (2013) as explained in Point 5. They reported that 100% MeOH did not allow the separation of the isoprostane isomers.
We have added the reason for using the methanol-acetonitrile phase to the manuscript as follows (line 172):
“Using methanol and acetonitrile mixture as the mobile phase (B) was based on previous work from Prasain et al. (2013) reporting that F2-isoprostane isomers separation was not achieved with a mobile phase of 100% methanol[29].”
Point 7: The anion exchange cartridges in SPE did not work well for 8-isoprostane. As the urine sample was acidified before SPE, did the Authors consider pH effect on SPE efficiency?
Response point 7:
We have specified in the text that the sample was basified before the loading on the anion exchange cartridge. The following sentence was added in the Supplementary Information:
“During the tests with anion exchange cartridges, the samples were adjusted to basic pH ranges with ammonium hydroxide (0.05%).”
Point 8: The retention time in LC, retention on SPE anion exchanger and MS ionization may suggest that it is worth extending the statement: "8-oxodG is composed of a purine and is a polar molecule due to the presence of polar functional groups (amides, hydroxyls, and amine). Moreover, it is uncharged under neutral pH and forms anions and dianions at higher pH values (pKa values at 8.6 and 11.7)" to lower pH situation.
Response point 8:
8-OxodG remains uncharged at low pH, and this is the reason for acidifying the samples. We added a sentence in line 358 as suggested by the reviewer.
“Moreover, it is uncharged under low and neutral pH and forms anions and dianions at higher pH values (pKa values at 8.6 and 11.7).”
Point 9: The chemical analytical techniques (line 453) could be specified as this phrase brings to mind a wet-technique analysis and derivatisation. (physicochemical, as mostly HPLC and MS are used?)
Response point 9:
The text was modified according to the review of Graille et al. (2020), line 401.
“For 8-oxodG, the authors reported a median value obtained by chromatographic analytical techniques in healthy adults of 3.9 ng/mg creatinine (3 – 5.5 ng/mg creatinine) with a BMI ≤ 25 and 2.8 ng/mg creatinine (2.4 – 3.5 ng/mg creatinine) with a BMI > 25.”
Point 10: In conclusion, I find this text really interesting and full of useful information, however, the details require careful reading to understand the significance of experimental design.
Response point 10:
We thank the reviewer for the positive comments. We moved technical details in Supplementary Information to simplify the text for readers not too familiar with analytical chemistry.
Point 11: Comments on the text preparation: There are inconsistencies between the text and data in table 3.
Response point 11:
We thank the reviewer for detecting this inconsistency. The text have been modified to correspond to Table 2 (line 196):
“Fifty-six participants provided first-void urine samples for the quantification of 8-oxodG and 8 isoprostane. Mean age of the participants was 43.5 years old and a BMI mean of 26. Twenty-six participants were women (46%) and 30 participants men (54%). Participant demographics are described in Table 2.”
Point 12: In the sentence there seem to be a redundancy: (192-193) Criteria for linearity was a coefficient of determination for urinary calibration curve greater than R2>0.999.
Response point 12:
The sentence was modified to remove the redundancy as follows (in the Supplementary Information):
“Criteria for linearity was a coefficient of determination R2 greater than 0.999 for the urinary calibration curve.”
Point 13: There are fragments of the text that may benefit from re-wording: (52) 8-oxodG is one of the major lesions resulting from oxidative damage to DNA
Response point 13:
The text in the manuscript has been rephrased as:
“8-oxodG is one of the major compounds resulting from oxidative damage to DNA.”
Point 14: (72-73) Urine is the preferred matrix in biological monitoring because it is easily obtainable and a noninvasive sampling method.
Response point 14:
The manuscript text has been rephrased as suggested by the reviewer:
“Urine is the preferred matrix in biological monitoring because its collection involves a simple, non-invasive sampling method.”
Point 15: Plural forms in several fragments of the text seem strange: (211) urines samples were
Response point 15:
We have corrected this throughout the text.
Reviewer 3 Report
Title “Rapid liquid chromatography – tandem mass spectrometry analysis of two urinary oxidative stress biomarkers: 8-oxodG and 8-isoprostane”, authors Nicolas Sambiagio, Jean-Jacques Sauvain, Aurélie Berthet et al.
In this article, the authors present the oxidative stress on urinary effect and analytical methods for the determination of biomarkers such as 8-oxo- 7,8-dihydro-2’-deoxyguanosine (8-oxodG) and 8-iso-prostaglandin F2α (8-isoprostane). Using of rapid ultra-high-performance liquid chromatography with a reversed-phase column and a gradient consisting of 0.1% acetic acid in water and 0.1% acetic acid in methanol/acetonitrile (70:30), which is connected to tandem mass spectrometry (UPLC-MS/MS) allow the amount of both urinary biomarkers to be determined.
The results are very interesting and this newly developed method is applicable for biomonitoring studies as well as large pidemiological studies on the effect of oxidative damage.
56 urine samples from healthy non-smoking participants were successfully analyzed and the amount of background levels of biomarkers of oxidative stress was presented.
Author Response
Comments Reviewer 3
Point 16: Title “Rapid liquid chromatography – tandem mass spectrometry analysis of two urinary oxidative stress biomarkers: 8-oxodG and 8-isoprostane”, authors Nicolas Sambiagio, Jean-Jacques Sauvain, Aurélie Berthet et al. In this article, the authors present the oxidative stress on urinary effect and analytical methods for the determination of biomarkers such as 8-oxo- 7,8-dihydro-2’-deoxyguanosine (8-oxodG) and 8-iso-prostaglandin F2α (8-isoprostane). Using of rapid ultra-high-performance liquid chromatography with a reversed-phase column and a gradient consisting of 0.1% acetic acid in water and 0.1% acetic acid in methanol/acetonitrile (70:30), which is connected to tandem mass spectrometry (UPLC-MS/MS) allow the amount of both urinary biomarkers to be determined. The results are very interesting and this newly developed method is applicable for biomonitoring studies as well as large epidemiological studies on the effect of oxidative damage. 56 urine samples from healthy non-smoking participants were successfully analyzed and the amount of background levels of biomarkers of oxidative stress was presented.
Response point 16:
We thank the reviewer for the positive comments on the manuscript and for highlighting the usefulness of the method for biomonitoring studies.
Round 2
Reviewer 1 Report
The supplementary information brings the manuscript closer to the journal topic.
Analytically, the manuscript is perfect.